# Position: What Cézanne Knew About Visual Intelligence That Vision-Language Models Miss

Mohammad Rashedul Hasan [1]   Chinh Hoang [1]

## Abstract

This position paper argues that vision-language model benchmarks for causal reasoning contain two blind spots. First, benchmarks presuppose temporal constitution, the understanding of time as the medium through which causes bring about effects, without testing it. Second, they provide scaffolding through prompts that may substitute for internalized capability instead of accessing it. A camera records one instant from one position. Human visual intelligence works differently. Post-impressionist painter Paul Cézanne grasped this a century before cognitive science would articulate it. He painted how we see, not what is seen. We draw on his technique to distinguish camera-like processing from Cézanne-like construction of understanding through temporal integration. Preliminary evidence shows systematic disparity between fluent causal text and valid causal structure, and divergent responses to identical scaffolding manipulation. Progress requires benchmarks that go beyond output accuracy to test temporal understanding and scaffolding-invariance.

## 1. Introduction

Vision-language models (VLMs) are being deployed in high-stakes applications where causal reasoning determines outcomes. In medical imaging, VLMs generate diagnostic reports and answer clinical questions about radiographs, pathology slides, and CT scans (Hartsock & Rasool, 2024; Chen et al., 2024c). In autonomous driving, VLMs perform scene understanding, behavior prediction, and trajectory planning (Zhou et al., 2024; Tian et al., 2024). These applications require models to reason about what would happen

[1]Department of Electrical and Computer Engineering, University of Nebraska-Lincoln, Nebraska, USA. Correspondence to: Mohammad Rashedul Hasan <hasan@unl.edu>.

*Proceedings of the 43rd International Conference on Machine Learning*, Seoul, South Korea. PMLR 306, 2026. Copyright 2026 by the author(s).

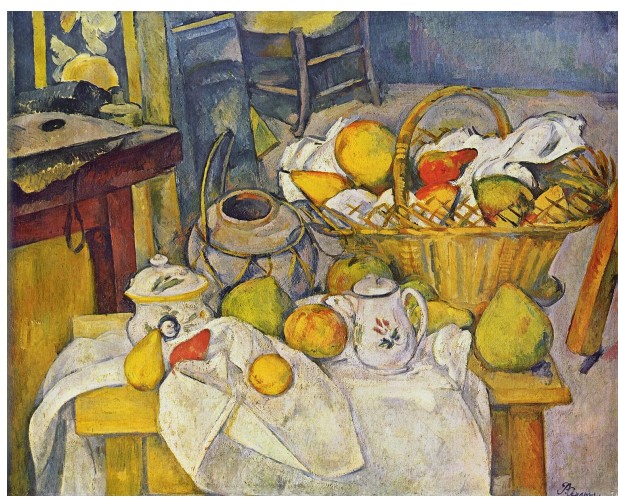

*(a)* Cézanne's painting

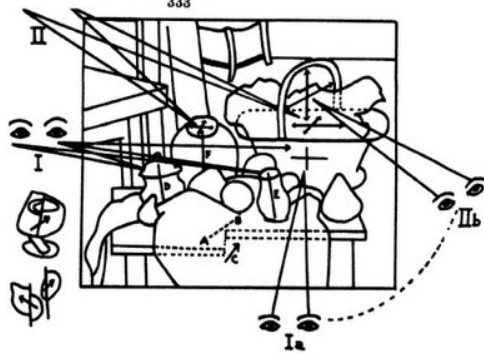

*(b)* Geometric analysis

*Figure 1.* Paul Cézanne, *Kitchen Table* (1888–90). No photograph could capture this composition. Geometric analysis (Loran, 2006) shows that each object is rendered from a separate vantage point. Positions I, II, Ia, and IIb mark four distinct eye levels. This is the accumulation of looking. A camera records one instant from one position. Human visual intelligence works differently. We build understanding by integrating shifting viewpoints over time instead of extracting from a single frozen instant. Cézanne grasped this a century before cognitive science would articulate it. He painted how we see, not what is seen from a frozen position. This distinction grounds our critique of VLM causal reasoning evaluation. Section 2 develops the framework.

under different conditions, such as if a lesion were larger, a vehicle changed lanes, or medication were administered. The stakes of misdiagnosis or collision make it urgent to un-

derstand whether VLMs possess genuine causal reasoning or generate plausible-sounding outputs through other means. The urgency intensifies as VLMs move from research benchmarks to clinical and safety-critical deployment, where evaluation gaps become deployment failures.

Benchmarks for evaluating causal reasoning in VLMs have proliferated accordingly. Researchers have developed evaluations targeting each level of Judea Pearl's Ladder of Causation (Pearl, 2009; Pearl & Mackenzie, 2018). These span association (Level 1, observing correlations), intervention (Level 2, reasoning about action effects), and counterfactual reasoning (Level 3, imagining alternative scenarios) (Yi et al., 2020; Chen et al., 2024b; Li et al., 2025; Zhang et al., 2024; Komanduri et al., 2025; Liu et al., 2025). Recent work has extended evaluation to temporal dimensions, testing whether models can infer temporal ordering from static visual markers such as rust or decay (Wang et al., 2025).

A persistent gap, however, separates benchmark performance from genuine causal understanding. Findings from temporal causality benchmarks illustrate this gap. When temporal order is provided explicitly through prompting, models perform reasonably well by drawing on commonsense knowledge. When they are required to infer temporal order without such prompting, performance degrades substantially (Wang et al., 2025). Chain-of-thought evaluation exposes a related limitation. Given an image and candidate causal chains representing the causal structure of the depicted scene (e.g., "wind blows → umbrella tips → umbrella falls"), models struggle to select the valid chain despite the structural support that pre-provided candidates offer (Zhang et al., 2025b). Whether models can generate valid causal structures without such scaffolding is an open question.

These findings instantiate a distinction drawn in explainable AI research between *plausibility* and *faithfulness* in model outputs (Lyu et al., 2024; Agarwal et al., 2024). Plausible outputs seem logical and convincing to human evaluators. Faithful outputs accurately reflect structured reasoning processes. Recent work argues that these properties can diverge. Pressures to generate user-friendly outputs may increase plausibility without improving faithfulness (Agarwal et al., 2024), and the faithfulness of large language model (LLM) explanations varies substantially by task and model (Madsen et al., 2024). This **plausibility-faithfulness gap** poses particular risks in high-stakes domains where understanding the reasoning behind a decision is as consequential as the decision.

The plausibility-faithfulness gap pervades VLM evaluation. Models confidently generate plausible explanations and ignore contradictory visual evidence (Guan et al., 2024), generate hallucinated explanations for straightforward visual questions (Tong et al., 2024), struggle with structured causal

reasoning despite fluent verbalization (Chen et al., 2024b), and rely on spurious correlations instead of genuine visual understanding (Yang et al., 2025). These findings and the scaffolding dependence described above indicate that models verbalize plausible causal narratives through pattern associations but do not construct structured temporal-causal representations (Zečević et al., 2023). **Apparent capabilities depend on external support.**

What underlies this dependence? One interpretation treats it as a technical limitation addressable through improved training (Xu et al., 2025; Zhang et al., 2025c), prompt design choices (Chen et al., 2024b), or larger and more diverse pretraining corpora (Chen et al., 2024a). On this view, VLM shortcomings primarily reflect data or optimization gaps. We propose a different interpretation. **The gap persists because current benchmarks evaluate reasoning outputs while presupposing temporal-causal foundations they do not examine.** The temporal causality findings show this. Benchmarks evaluate inference about temporal ordering from static visual cues (such as rust or decay), while leaving unspecified whether models possess what we call *temporal constitution*, the understanding of time as the medium through which causes bring about effects. Inferring that one state preceded another differs from representing duration, process, and irreversibility as foundations for reasoning. When such foundations are assumed or externally scaffolded, models can generate plausible explanations. When they are absent, performance degrades. This suggests a limitation tied to representational capacity and evaluation design, not only to training scale.

How might temporal constitution manifest in visual understanding? A camera records one instant from one position. Human visual intelligence works differently. We build understanding by integrating shifting viewpoints over time instead of extracting from a single frozen instant. Post-impressionist painter Paul Cézanne grasped this a century before cognitive science would articulate it. His canvases render multiple viewpoints within a single composition (Figure 1). They show how visual understanding is built, not what is seen from a frozen position. Current VLM architectures inherit the camera's paradigm. Our diagnostic framework tests whether single-frame feature extraction yields the temporal encoding that genuine causal reasoning requires. Section 2 develops this distinction into a theoretical framework with testable diagnostic implications.

**This position paper argues that VLM causal reasoning evaluation contains two blind spots that substantially limit the diagnostic value of current benchmarks.** *First*, existing evaluations test counterfactual reasoning while assuming that models already understand time without distinguishing time-as-data (temporal information extracted from visual inputs) from time-as-constitution (the temporal

depth through which understanding develops). *Second*, evaluations provide scaffolding through prompts and question structure while assuming this helps models access internalized knowledge without recognizing the distinction between external scaffolding and constitutive internalization. These blind spots help explain why the plausibility-faithfulness gap persists despite architectural advances. We evaluate outputs without probing whether models possess the temporal and symbolic foundations that genuine causal reasoning requires.

## 2. Theoretical Framework and Hypotheses

The two blind spots identified in Section 1 have theoretical grounding in art, philosophy, and psychoanalysis. These disciplines identify distinctions that current AI evaluation overlooks. *Temporal constitution is the prerequisite foundation*; *scaffolding-invariance is the diagnostic test* for whether that foundation has been achieved.

**The Temporal Blind Spot: A Prerequisite Foundation.** Causal reasoning is inherently temporal. Effects do not precede their causes. Counterfactual reasoning requires mentally simulating how alternative pasts would have led to different presents (Sloman, 2005; Sloman et al., 2009). This temporal structure is not incidental to causal understanding but constitutive of it. Without encoding time as the medium through which causes bring about effects, a system cannot reason about what would have happened otherwise.

What does this mean concretely for VLMs? A VLM shown an image of a rusted bicycle, when prompted about why the bicycle looks this way, outputs "This bicycle is old because of the rust." The same VLM shown a cracked sidewalk next to a tree, when prompted about the cause of the damage, outputs "The tree roots caused the crack." Both answers are plausible and likely correct. But different processes could have generated these outputs.

The first process extracts statistical associations from frozen visual features. Rust co-occurs with age. Trees near cracks co-occur with root damage. The model pattern-matches these associations from training data. The second process constructs a representation of temporal unfolding. Rust accumulates through ongoing, irreversible oxidation. The bicycle's current state constrains its possible futures. Roots grow over years, exert gradual pressure, and stress accumulates until the fracture threshold is reached. Both processes can generate correct outputs for familiar scenes. Only the second reflects genuine causal understanding that would generalize to novel scenarios. Current benchmarks measure whether outputs are correct. They cannot distinguish extraction from construction.

What does constructed understanding look like in visual perception? Section 1 introduced the distinction be-

tween camera-like extraction and human visual intelligence. Cézanne's *Kitchen Table* still life (Figure 1a) shows this distinction in action.

When we examine a complex scene, we shift position, lean in, refocus on different objects. The brain weaves these moments into coherent understanding. Cézanne painted this synthetic process. In the *Kitchen Table*, objects tilt in different directions. The table's left edge slopes one way, the right edge another. The ginger jar is viewed from above while the basket tilts toward the viewer. Geometric analysis identifies four distinct eye levels (Figure 1b) (Loran, 2006). Positions I, II, Ia, and IIb mark different viewing angles, with sight lines showing each object rendered from a separate vantage point. No single photograph could capture this composition. The painting encodes different bodily relationships to space. The tabletop is arm's-length space where hands reach and arrange, while the tipped-up floor is standing space where feet bear weight. The background is looking-across space. When we stand at a table, we are aware of all three at once. It is the accumulation of looking. Multiple moments of perception integrated into one coherent representation. Phenomenological analysis articulates this understanding (Merleau-Ponty, 1964). Geometric perspective represents the world as seen by a disembodied eye at a single instant. Cézanne rendered perception as it is lived through embodied and temporal engagement.

We call camera-like extraction and Cézanne-like construction the two poles of this distinction. This gap defines temporal constitution in visual understanding. A system capable of genuine causal reasoning would require temporal encoding analogous to what human perception achieves. Such encoding represents that causes precede effects, that processes unfold with duration, and that present states constrain future possibilities. Current VLM architectures process images as cameras do, extracting features from single frozen moments. We propose a diagnostic framework to test whether any amount of such single-frame feature extraction can yield this temporally integrated understanding.

The contrast between camera-like and Cézanne-like processing has computational grounding in predictive processing theory. The brain actively generates hypotheses and continuously updates internal models through iterative engagement with the world (Friston, 2010; Seth, 2021). Understanding is built through cycles of prediction and error correction, not instantaneous feature extraction.

David Hume's analysis of causal inference sharpens the concern that pattern matching may masquerade as causal understanding (Hume, 2008). We never perceive causation, only sequence. Causal understanding requires counterfactual imagination. If the first had not been, the second never had existed. Pattern matching to observed sequences differs from the temporal-imaginative capacity that grounds

genuine causal reasoning. A model can learn that rust co-occurs with old bicycles without representing oxidation as a temporal process. A model can learn that trees near cracks co-occur with root damage yet never encode the years of gradual pressure that led to fracture. Temporal inference from static markers is not the same as temporal constitution.

Video-language model research provides empirical evidence that pattern matching can succeed on causal benchmarks. Single-frame models that ignore temporal information achieve competitive or superior performance on tasks designed for temporal and causal reasoning (Buch et al., 2022; Lei et al., 2023). Models succeed by recognizing visual patterns associated with outcomes but do not represent the temporal process that connects cause to effect.

Current VLM benchmarks test whether models generate correct causal outputs but not whether they possess the temporal-constructive capacity that underlies genuine understanding. The temporal blind spot is a *prerequisite* problem. Without temporal constitution, genuine causal reasoning cannot occur, and the plausibility-faithfulness gap will persist regardless of training scale.

**The Symbolic Blind Spot: A Diagnostic Signature.** If temporal constitution is the prerequisite for genuine causal reasoning, how can we detect whether a model possesses it? French psychoanalyst Jacques Lacan's theory of symbolic constitution addresses this diagnostic question (Lacan et al., 2006). Symbolic structures encompass laws, norms, and systems of meaning that pre-exist individuals. Language is the primary medium of these structures. Its grammar and rules pre-exist any speaker. Humans do not learn language as an external tool. We are formed within it. Children do not acquire language as an external skill but become speakers by inhabiting linguistic structure that precedes them. A fluent speaker does not retrieve linguistic capacity from outside. The capacity is internal structure, part of the speaker's constitution. We apply Lacan's diagnostic logic to VLM causal reasoning. A model that has genuinely internalized causal frameworks should exhibit *scaffolding-invariance*. Genuine capability persists regardless of how it is externally prompted because the framework is internal to the model.

This distinction between external provision and constitutive internalization has diagnostic implications for VLM evaluation. VLMs are built on language model foundations and inherit their characteristic prompt sensitivity (Zhao et al., 2021; Lu et al., 2022; Sclar et al., 2024; Feng et al., 2025). Model outputs vary substantially with prompt wording, so practitioners optimize prompts to improve performance (Hoang et al., 2025). Existing work addresses how to find optimal prompts. Our question is different. Do sensitivity patterns expose the relation between model and causal framework? Sensitivity to scaffolding signals that the

framework remains an external dependency. When identical scaffolding manipulation generates opposite effects across models (one improves, one degrades), this pattern diagnoses absent constitutive internalization. Neither model inhabits causal frameworks from within. Each relates to them from outside, shaped by training history.

This diagnostic approach connects to longstanding debates about symbol grounding and systematic generalization in neural networks (Fodor & Pylyshyn, 1988; Lake & Baroni, 2018). Whether neural networks can achieve genuine symbolic competence or approximate it through statistical regularities is unresolved. Our contribution is translating this theoretical concern into testable diagnostics. We vary scaffolding systematically while holding task content constant. A model that has internalized causal reasoning as constitutive structure should show stable performance across scaffolding conditions. Scaffolding-invariance is a necessary but not sufficient condition for constitutive internalization. Stability could also arise from shallow heuristics that apply consistently. Sensitivity, by contrast, provides clear negative evidence. Current evaluation methodology cannot detect this pattern because it does not vary scaffolding systematically. Benchmark comparisons treat prompt structure as fixed infrastructure instead of as experimental variable. The question of constitutive internalization remains unexamined.

**The Hierarchical Relationship.** The two blind spots form a diagnostic hierarchy. Temporal constitution is the *prerequisite*. Genuine causal reasoning requires encoding time as the medium through which causes bring about effects. Scaffolding-invariance is the *diagnostic test* for whether temporal constitution has been achieved. A model lacking temporal foundation cannot exhibit scaffolding-invariance for causal reasoning because there is no internalized capability to remain stable. Scaffolding sensitivity provides evidence that causal frameworks remain external dependencies. This hierarchy explains why the plausibility-faithfulness gap persists. Models can generate plausible causal language through pattern matching while lacking the temporal constitution that would yield scaffolding-invariant structural reasoning. This hierarchy grounds our diagnostic framework.

**Hypotheses.** The framework motivates two hypotheses, summarized in Table 1.

*H1 (Temporal Prerequisite).* Models will show disparity between plausible causal text and valid causal structure. Recent work shows models struggle to select valid causal chains even when candidates are provided (Zhang et al., 2025b). We test the more demanding capability of generating valid chains without such scaffolding. High text fluency paired with poor chain validity would instantiate the plausibility-faithfulness gap. This pattern signals pattern matching without genuine causal construction.

*H2 (Symbolic Diagnostic).* Models will show sensitivity to scaffolding variation. Divergent responses to identical manipulation across models (one improves, one degrades) will show that no model has achieved constitutive internalization. Each relates to causal frameworks externally, shaped by its particular training.

*Table 1.* Hypotheses and their roles in the diagnostic hierarchy.

| Hypothesis | Role in Hierarchy | What It Tests | Diagnostic Metric |
|---|---|---|---|
| H1 (Temporal) | Prerequisite foundation | Whether models construct or pattern-match | Causal Depth Disparity |
| H2 (Symbolic) | Diagnostic signature | Whether capability is internalized | Scaffolding Sensitivity |

# 3. Diagnostic Framework and Preliminary Evidence

This section operationalizes H1 and H2 through diagnostic measures. We present proof-of-concept evidence from controlled experiments on three VLMs.

## 3.1. Operationalizing H1: Causal Depth Disparity

H1 predicts disparity between plausible causal text and valid causal structure. Building on recent work validating causal chains as a proxy for structural causal understanding (Zhang et al., 2025b), we test whether models can *generate* such chains, not just select valid chains from pre-provided candidates. Generation without scaffolding provides a more stringent test of internalized causal structure. We require models to generate both natural language explanations and explicit causal chains using arrow notation (e.g., *Strong wind → Force on umbrellas → Umbrellas topple*).

Chain specifications are deliberately minimal. Intervention questions (Pearl Level 2) require at least one causal link. Counterfactual questions (Level 3) require at least two. Failure to generate valid chains cannot be attributed to representational complexity. The notation requires only identifying causal elements and their sequential relationships. Chain validity serves as a minimal proxy for temporal-causal construction because links encode ordered dependence that counterfactual reasoning relies on.

**Definition 3.1** (Causal Depth Disparity). Let $M$ denote a VLM evaluated under scaffolding condition $c$, $\mathcal{D}$ a set of image-question pairs targeting causal reasoning, and let $T_c(M, d)$ and $G_c(M, d)$ denote scores for text responses and causal chain validity, respectively, for each $d \in \mathcal{D}$. The Causal Depth Disparity under condition $c$ is:

$$\mathrm{CDD}_c(M) = \frac{\mathbb{E}_{d \in \mathcal{D}}[T_c(M, d)] - \mathbb{E}_{d \in \mathcal{D}}[G_c(M, d)]}{\mathbb{E}_{d \in \mathcal{D}}[T_c(M, d)]} \quad (1)$$

CDD values near 1 signal high disparity (plausible text without valid chains). Values near 0 signal alignment. High CDD instantiates the plausibility-faithfulness gap. **It is the signature of camera-like processing.** Models pattern-match to causal language but cannot abstract causal structure. Our concurrent work on the Abstraction Gap provides converging evidence across multiple VLM architectures (Hoang & Hasan, 2026). The CDD metric operationalizes this disparity within a scaffolding-sensitivity framework. We test H2 by comparing CDD across scaffolding conditions.

## 3.2. Operationalizing H2: Scaffolding Sensitivity

H2 predicts sensitivity to scaffolding variation, with divergent patterns across models indicating absent constitutive internalization. We vary scaffolding systematically across three conditions.

- **Condition 1 (Full Scaffolding):** Detailed instructions, reference to Pearl's causal framework, worked examples, and output templates.
- **Condition 2 (Partial Scaffolding):** Detailed instructions and framework reference, but no worked examples.
- **Condition 3 (Minimal Scaffolding):** Brief task description with basic format indication only.

**Definition 3.2** (Scaffolding Sensitivity). For a VLM $M$, let $G_{c_1}(M)$ and $G_{c_3}(M)$ denote average causal chain validity under full scaffolding ($c_1$) and minimal scaffolding ($c_3$), respectively. Scaffolding Sensitivity is:

$$\mathrm{SS}(M) = G_{c_1}(M) - G_{c_3}(M) \quad (2)$$

Positive SS signals scaffolding dependence. Negative SS signals scaffolding interference. SS provides a coarse endpoint summary. Interpreting scaffolding response patterns requires examining the full condition-wise trajectory, as SS can obscure non-monotonic patterns. Divergent responses across models would support H2. Each model's relation to symbolic structure is shaped by training history instead of shared internalized capacity.

## 3.3. Evaluation Protocol

We test on 500 images from the COCO validation set (Lin et al., 2014), with 9 questions per image spanning Pearl's three causal levels (3 each for association, intervention, and counterfactual), yielding 4,500 items. Each model is tested under all three scaffolding conditions, generating 13,500 responses per model. Responses receive independent scoring from two automated judges (GPT-4o (OpenAI, 2024) and Claude 4.5 Sonnet (Anthropic, 2024)) using validated rubrics and following standard practice in VLM evaluation (Liu et al., 2024a). Text responses and causal chains

are scored separately on a 0–10 scale. Human review of a 30-image subset corroborates automated metrics. The Appendix provides complete prompts (Appendix A.1), scoring methodology (Appendix A.2), scoring prompts for automated judges (Appendix A.3), and detailed results with bootstrapped confidence intervals (Appendix A.4).

### 3.4. Model Selection

We compare three models from the LLaVA family that share architectural lineage but differ in training approach. **LLaVA-NeXT 13B** (Liu et al., 2024b) uses dynamic resolution with enriched instruction tuning. **LLaVA-RLHF 13B** (Sun et al., 2023) uses human feedback optimization with frozen image encoder. **LLaVA-CoT 11B** (Xu et al., 2025) uses chain-of-thought supervision targeting reasoning verbalization. This comparison isolates how training objectives shape models' relations to symbolic frameworks. Different training approaches should yield distinct scaffolding patterns.

### 3.5. Results and Analysis

**Evidence for H1: Causal Depth Disparity.** Table 2 presents CDD values for Level 3 (counterfactual) questions.

*Table 2.* Causal Depth Disparity (CDD) for Level 3 counterfactual questions. Values near 0 indicate text-chain alignment. Values near 1 indicate high disparity. [†]LLaVA-RLHF Condition 1 generated 2,467 blank responses, yielding poor scores on both metrics (Text=1.92, Chain=1.11). Detailed results in Appendix A.4.

| Model | Full (C1) | Partial (C2) | Minimal (C3) |
|---|---|---|---|
| LLaVA-NeXT 13B | 0.00 | 0.07 | 0.16 |
| LLaVA-RLHF 13B[†] | 0.30 | 0.90 | 0.87 |
| LLaVA-CoT 11B | 0.76 | 0.86 | 0.66 |

LLaVA-NeXT illustrates the diagnostic value of our hierarchical framework. Under full scaffolding, it achieves near-zero CDD (text: 7.85, chain: 7.42), the strongest text-chain alignment among tested models. LLaVA-NeXT differs from other LLaVA variants in its AnyRes dynamic resolution handling and enriched instruction tuning mixture (Liu et al., 2024b). Standard evaluation under fixed scaffolding would conclude that LLaVA-NeXT possesses robust causal reasoning capability. Varying scaffolding tells a different story. CDD increases systematically (0.00 → 0.07 → 0.16), and chain scores decline 25% (7.42 → 5.59). H1 (text-chain alignment) can be satisfied under scaffolded conditions while H2 (scaffolding-invariance) fails. Without scaffolding variation, this dependence stays invisible.

LLaVA-CoT shows substantial CDD (0.66–0.86) despite explicit training on reasoning chains. Text scores reach 7.16 but chain scores only 1.34 under full scaffolding. This supports H1. Verbalization and abstraction are distinct capabilities. LLaVA-RLHF shows an anomalous pattern where full scaffolding generates 2,467 blank responses. Removing worked examples (C2) recovers text scores (6.39) while

chain scores remain poor (0.47), spiking CDD to 0.90.

**Evidence for H2: Scaffolding Divergence.** Table 3 presents chain quality and Scaffolding Sensitivity.

*Table 3.* Chain quality (0–10) for Level 3 questions with Scaffolding Sensitivity (SS = $G_{c_1} - G_{c_3}$). Positive SS indicates dependence. Negative SS indicates interference. [†]LLaVA-RLHF C1 generated 2,467 blanks. SS obscures this pattern.

| Model | Full (C1) | Partial (C2) | Minimal (C3) | SS |
|---|---|---|---|---|
| LLaVA-NeXT 13B | 7.42 | 6.31 | 5.59 | +1.83 |
| LLaVA-RLHF 13B[†] | 1.11 | 0.47 | 0.92 | +0.19 |
| LLaVA-CoT 11B | 1.34 | 0.97 | 2.41 | −1.07 |

The three models exhibit divergent scaffolding responses. This supports H2. LLaVA-NeXT shows positive SS (+1.83). Chain quality degrades monotonically as scaffolding is removed. LLaVA-CoT shows negative SS (−1.07). Chain quality improves when scaffolding is minimized. LLaVA-RLHF shows a qualitatively different pattern that the endpoint-based SS metric obscures. Full scaffolding causes catastrophic failure, partial scaffolding causes further degradation, and minimal scaffolding allows partial recovery.

**Three Patterns of External Dependence.** The divergent responses instantiate three qualitatively different relationships to scaffolding.

- **LLaVA-NeXT** (SS = +1.83): Scaffolding as *necessary support*. Valid chains require external symbolic structure. Capability degrades when scaffolding is removed.
- **LLaVA-CoT** (SS = −1.07): Scaffolding as *interference*. CoT training patterns conflict with external scaffolding. Removing it improves (though does not fully restore) chain generation.
- **LLaVA-RLHF** (SS = +0.19): Scaffolding as *format violation*. RLHF-shaped expectations conflict with our scaffolding format, causing catastrophic output failure.

All three patterns indicate external, contingent relations to symbolic structure shaped by training history. The observed divergence suggests that none of the tested models inhabits the causal reasoning framework as internal structure. **All three remain in camera-like processing mode**. They extract statistical associations from frozen features. None constructs causal understanding through temporal integration.

The CDD and SS patterns support the hierarchical relationship between H1 and H2. LLaVA-CoT illustrates this most clearly. High CDD (text fluency without chain validity) paired with negative SS (scaffolding interference) shows that CoT training improved verbalization but did not instill temporal-constructive capacity for causal abstraction. Verbalization and abstraction are distinct capabilities.

**Scope and Limitations.** These results derive from three VLMs within the LLaVA family. We do not claim generalization across all architectures. Our goal is proof-of-concept

evidence that current evaluation practices may miss genuine capability deficits. The controlled comparison within a single model family allows differences in scaffolding response to be attributed to training approach, not architectural variation. Whether similar patterns appear in other model families remains an empirical question. The LLaVA-RLHF results also illustrate that evaluation formats can interact with training objectives in unexpected ways, reinforcing our central point. Models' relations to symbolic structure are contingent on training history and evaluation format. Benchmarks that hold format constant cannot detect these dependencies.

## 4. Call to Action

Our preliminary findings have practical implications. High-stakes applications require genuine causal understanding. Systems that generate plausible explanations without valid causal structure may fail unpredictably in deployment.

Our diagnostic framework addresses a specific instance of a broader validity crisis in AI evaluation. Output-based benchmarks face systematic challenges (Bowman & Dahl, 2021; Raji et al., 2021). Recent work argues that GenAI evaluation is a social science measurement challenge that requires rigorous validity frameworks (Wallach et al., 2025). Our scaffolding variation methodology complements these approaches by testing whether capability depends on external symbolic support. This section proposes directions for benchmark developers and VLM researchers, and identifies criteria for evaluating causal reasoning claims.

**For Benchmark Developers.** Current benchmarks test counterfactual outputs under fixed scaffolding. We propose three directions for extension, intended to augment existing benchmarks, not replace them.

*Test Temporal Foundations.* Before testing counterfactual reasoning, probe temporal understanding as a prerequisite. Diagnostic evaluation could assess whether models correctly sequence cause-effect relationships when temporal markers are ambiguous, distinguish reversible from irreversible processes, identify how changes propagate across different time scales, and trace how alternative pasts would have led to different presents. Models that fail such temporal probes should not receive full credit for correct counterfactual answers. Such answers may result from camera-like pattern matching instead of Cézanne-like temporal construction.

*Probe Causal Structural Understanding.* Benchmarks should not rely solely on linguistic fluency to assess causal reasoning. Our dual-probe approach (requiring both text explanations and causal chains) shows one method for testing whether models can externalize causal structure, not just verbalize causal narratives. We offer this approach as

illustrative. Future work should investigate more effective probes of structural understanding. Chain generation is a minimal proxy. Richer structural representations may provide greater diagnostic value.

*Vary Scaffolding Systematically.* Scaffolding encompasses more than few-shot versus zero-shot prompting. It includes theoretical framing (e.g., explicit reference to Pearl's causal hierarchy), worked examples, output templates, and detailed task instructions. Current VLM benchmarks often employ substantial scaffolding but do not report sensitivity to its variation (Sclar et al., 2024; Feng et al., 2025). Evaluation should span multiple scaffolding conditions, from full theoretical framing with worked examples to minimal task descriptions, and report performance across all conditions. A model's score under fixed scaffolding reflects its relationship to that scaffolding, not underlying capability. Dynamic benchmarks with procedurally generated content address memorization concerns (Jin et al., 2024). Combining dynamic content generation with systematic scaffolding variation would provide a more complete diagnostic picture.

**For VLM Researchers.** Current training approaches generate different relationships to symbolic structure, but none in our experiments yielded scaffolding-invariance. Recent work shows why. Counterfactual fine-tuning with specialized loss functions improves similarity-based discrimination on compositional reasoning benchmarks but does not transfer to structural generation tasks (Zhang et al., 2025a). Chain-of-thought supervision improves verbalization fluency but, as our evidence suggests, does not yield corresponding gains in causal chain validity. These approaches optimize for discrimination in continuous embedding space. Scaffolding-invariant causal reasoning may require discrete structural abstraction that current objectives do not target.

We identify three directions that address this gap.

*Structure-First Training Objectives.* Current training generates natural language and hopes causal structure emerges implicitly. An alternative is to train models to predict causal structure explicitly, with natural language as a secondary output. This might involve auxiliary losses for causal chain prediction that penalize structural invalidity independent of text fluency, multi-task learning that jointly optimizes text quality and structural validity with separate supervision signals, or curricula that require structural validity before rewarding fluency. The hypothesis is that explicit structural supervision may yield representations that support scaffolding-invariant reasoning.

*Scaffolding Curriculum.* Models trained under fixed scaffolding conditions learn to depend on that scaffolding. Training under systematically varied scaffolding, from full support to minimal prompts, may discourage this dependence. A scaffolding curriculum would progressively reduce

external structure (removing worked examples, then theoretical framing, then detailed instructions) and require stable performance throughout. Models that maintain performance as scaffolding decreases would signal internalization.

*Temporal Encoding Mechanisms.* Current architectures encode temporal information implicitly through sequential frame processing or positional embeddings. Given that temporal constitution is a prerequisite for causal reasoning (H1), architectures that explicitly represent duration, ordering, and process may perform differently. This might involve temporal attention mechanisms that attend across time scales or training objectives that predict temporal structure (ordering, duration, reversibility) from static images. Whether such mechanisms would yield genuine temporal constitution or improved temporal inference remains an empirical question our diagnostic framework could help answer.

These directions are speculative. We do not claim they will succeed, but they follow from our diagnosis and offer testable hypotheses for future work.

**Evaluating Causal Reasoning Claims.** Claims of causal reasoning capability are strengthened by evidence beyond single-condition accuracy.

*Scaffolding Variation.* Performance under fixed scaffolding may not generalize. Strong claims would report performance across multiple scaffolding conditions (varying theoretical framing, examples, and instruction detail) or explicitly acknowledge the scope limitation.

*Structural Validity.* High text fluency can mask absent causal structure. Strong claims would separate text quality from structural validity. Improvements should show up in structural outputs, not only linguistic fluency. This separation helps distinguish Cézanne-like construction from camera-like pattern matching.

*Generalization Evidence.* Strong performance under evaluation conditions that match training scaffolding does not guarantee deployment generalization. Strong claims would provide evidence that capabilities persist under varied conditions.

These criteria apply to our own preliminary evidence, which we present as proof-of-concept within a single model family, not as broad evaluation.

## 5. Alternative Views

The claims above invite reasonable skepticism. We address six objections that challenge our framework's premises, methodology, or practical relevance.

**Scaling Will Solve This.** One might argue that temporal constitution and symbolic internalization will emerge at sufficient scale. LLMs show emergent capabilities with scale (Wei et al., 2022; Kaplan et al., 2020). Perhaps causal understanding will similarly emerge in larger VLMs.

We are skeptical. Scaling provides more data and parameters but does not alter the relationship between model and training signal. A model trained on one billion images still processes each as an isolated snapshot. This is camera-like processing at scale, not a transition to Cézanne-like construction. The model does not build understanding through sustained engagement with any subject. Systematic evaluations support this concern. State-of-the-art models show significant limitations on counterfactual tasks despite massive scale (Zhang et al., 2023; Kıcıman et al., 2024), and recent work shows that scaling improves surface fluency faster than structured reasoning capability (McKenzie et al., 2024).

Even sophisticated training approaches designed for causal reasoning do not close the gap. Counterfactual fine-tuning with specialized loss functions improves compositional reasoning on similarity-based benchmarks but does not transfer to structural generation tasks (Zhang et al., 2025a). Our evidence (Section 3) points in the same direction. LLaVA-CoT and LLaVA-RLHF received targeted training (explicit reasoning supervision and human feedback optimization, respectively), but neither yielded scaffolding-invariance or low disparity between text fluency and chain validity. Targeted training specifically designed for counterfactual reasoning does not yield constitutive internalization. The assumption that untargeted scaling will do so lacks clear motivation.

**Video Models Address Temporal Understanding.** Video-language models process temporal sequences and might possess the temporal constitution we describe. Image-based VLMs lack temporal data. Video models might address this limitation.

This objection conflates time-as-data with time-as-constitution (Section 2). Video models process sequences of frames and receive temporal data. But processing frames sequentially does not entail understanding time as the medium through which causal relationships unfold. Sequential frame processing is camera-like processing applied to multiple moments, not Cézanne-like integration of temporal experience into coherent understanding. Research on video-language models finds strong "static appearance bias." Single-frame models achieve competitive or superior performance on tasks designed for temporal and causal reasoning (Buch et al., 2022; Lei et al., 2023). Questions labeled "causal" can often be answered from scene recognition in a single frame.

Testing video-language models with the type of diagnostics we present (dual outputs, scaffolding variation) would be valuable future work. We hypothesize similar patterns but remain open to evidence that video training yields quali-

tatively different temporal capacity. Evidence that video training yields scaffolding-invariant causal structure would count against our position.

**This Is Just Prompt Sensitivity.** Scaffolding sensitivity is well-established (Zhao et al., 2021; Lu et al., 2022; Sclar et al., 2024; Feng et al., 2025). Our experiments may rediscover this known phenomenon without adding insight.

Our contribution is not discovering sensitivity but using it diagnostically. The three-model comparison goes beyond generic prompt sensitivity. Identical manipulation generates qualitatively different effects (necessary support, format violation, interference). Generic prompt sensitivity would produce consistent effects across models. Divergent responses suggest that no tested model inhabits causal frameworks constitutively. Each relates to symbolic structure from outside in ways shaped by training history.

Prompt optimization addresses symptoms. Our diagnostic approach distinguishes scaffolding-dependent performance from genuine capability.

**Pattern Matching May Suffice.** One might argue that genuine causal understanding is unnecessary for deployment. Models generate correct answers. Why should we care whether they achieve this through pattern matching?

This objection has merit for constrained applications where test conditions match training. The concern arises for novel scenarios and high-stakes applications. Measuring genuine capability requires going beyond accuracy to examine the processes underlying performance (Chollet, 2019). Camera-like pattern matching degrades when conditions shift beyond training distribution (Marcus, 2020). A model without Cézanne-like causal construction cannot reason about unseen scenarios or distinguish causal factors from spurious correlates.

Diagnostics that measure disparity between text fluency and structural validity address this concern. When models generate fluent causal explanations but poor causal chains, this asymmetry signals that correct text may emerge from pattern matching. Structural understanding that would support generalization is absent. For medical reasoning or autonomous systems, this disparity undermines confidence in evaluation-based claims. Text-only evaluation misses this warning signal.

**Chain Validity Is an Imperfect Proxy.** One might object that chain validity does not capture temporal-causal understanding. A model could generate valid chains through template matching without causal construction, or possess causal understanding that our chain format fails to elicit.

We acknowledge this limitation. Chain validity serves as a minimal proxy, not a complete measure. The diagnostic value lies in disparity, not absolute chain scores. When

models generate fluent causal explanations but cannot generate even minimal valid chains, this asymmetry is informative regardless of whether chains perfectly capture causal understanding. The proxy is imperfect, but the pattern it detects (high text fluency, poor structural validity) would be difficult to explain if models possessed genuine causal construction capacity. Future work should develop richer structural probes. Our contribution is establishing that such probes are necessary.

**Cross-Domain Frameworks Are Inappropriate.** Importing frameworks from art history, philosophy, and psychoanalysis into AI research may be methodologically inappropriate. These domains explain human phenomena. Applying them to artificial systems may constitute category error.

We do not claim VLMs should function as human perception or undergo human development. The frameworks serve a different purpose. They identify distinctions that current AI evaluation overlooks. Interdisciplinary approaches have proven productive, from cognitive science informing architectures (Collins et al., 2024) to philosophy informing alignment (Gabriel, 2020).

The distinctions we draw (momentary extraction vs. constructed understanding, external provision vs. constitutive internalization) apply to any system claiming causal reasoning, regardless of mechanism. The frameworks are justified by diagnostic utility. They generated hypotheses that detected patterns (divergent scaffolding responses, high text-chain disparity despite reasoning training) that standard evaluation would miss.

## 6. Conclusion

The implication of our analysis is not that current VLMs lack value, but that current evaluation practices may overstate what their success signifies. Benchmark leaderboards may be tracking scaffolding optimization instead of causal reasoning capability. Deployment decisions rest on unexamined assumptions.

The deeper question is architectural. Current VLMs inherit the camera's frozen-moment paradigm. They process images as cameras do and extract features from isolated instants. Whether this paradigm can ever yield genuine causal understanding, or whether new architectures are needed, remains open. Human visual intelligence operates on different principles. We build understanding through sustained temporal engagement, not from frozen instants. Cézanne intuited this a century before cognitive science would formalize it. He painted how we see, not what is seen. AI has not yet grappled with what he discovered. The plausibility-faithfulness gap will persist until evaluation shifts from outputs to foundations and distinguishes camera-like extraction from Cézanne-like construction.

## Acknowledgments

This research was supported by a standard grant from the U.S. National Science Foundation (NSF DUE 2142558).

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

# A. Appendix

In this appendix, we provide supplementary material including complete prompts used for all three instruction conditions A.1, a description of the evaluation and scoring methodology A.2, the evaluation prompt for automated judges A.3, and detailed result tables with confidence intervals via boostraping A.4.

## A.1. Complete Instruction Prompts

We present the complete prompts used for all three instruction conditions. The progression systematically removes scaffolding components to test instruction dependence: Condition 1 provides maximum support (examples + detailed instructions), Condition 2 removes examples while retaining instructions, and Condition 3 provides only a minimal task description. All prompts reference Pearl's framework and request chains first, followed by text explanations. The placeholder {} indicates where the specific question is inserted.

### A.1.1. CONDITION 1: FULL SCAFFOLDING

This condition provides maximum scaffolding, including three worked examples, detailed step-by-step instructions, explicit Pearl framework references, and format templates.

---

**Condition 1 Prompt**

**System:** You are a vision-language model (VLM) tasked with answering a provided question about an image in a way that demonstrates causal reasoning, based on Judea Pearl's framework. You are encouraged to construct a lightweight causal chain to represent the chain of cause-and-effect relationships relevant to the answer, using the number of nodes and arrows needed to accurately capture the causal logic (e.g., 2 nodes like A → B, 3 nodes like A → B → C, or more if appropriate), tailored to the complexity of the reasoning. However, it is reasonable to provide only a text response without a causal chain if you find it sufficient to explain the causal reasoning. If you include a chain, provide it first, followed by a concise text response that explains the reasoning based on the chain. If you omit the chain, provide only the text response, ensuring it still reflects causal reasoning.
**Instructions:**
- **Base the response on the image and question:** Ensure the answer (and chain, if provided) is grounded in the image's content and relevant to the question.
- **Optional causal chain:** If you choose to include a chain, follow these steps:

  1. Identify the key cause-and-effect relationships implied by the question (e.g., 'Rain causes wet ground, which darkens the scene').
  2. Format these relationships as a chain using '→' to connect nodes (e.g., Rain → Wet Ground → Darkened Scene).
  3. Use the fewest nodes needed for clarity, but include additional nodes if the causal chain requires multiple steps (e.g., 2, 3, or more nodes).

- **Provide a concise answer:** Write 1–2 sentences that explain the causal reasoning, referencing the chain's steps if a chain is provided, or directly addressing the question's causal logic if no chain is included.
- **Avoid overly simplistic chains:** Ensure the response (and chain, if provided) captures specific, relevant causal relationships (e.g., avoid vague terms like 'Change → Outcome' unless fully justified).
- **Format the output exactly as follows:**

  If including a chain:

  ```
  Causal Chain:  [Your causal chain, e.g., A → B or A → B → C → D]
  Answer:  [Your answer, explaining the reasoning based on the chain]
  ```

  If omitting the chain:

  ```
  Answer:  [Your answer, explaining the causal reasoning]
  ```
**Examples:**
*Question:* How would the scene change if a sudden rainstorm began in a sunny park?
```
Causal Chain:  Rainstorm → Wet Ground → Darkened Atmosphere
Answer:  The rainstorm wets the ground, creating a darker, moodier atmosphere as
rain falls.
```
*Question:* What would the park look like if it had been photographed at night instead of midday?
```
Causal Chain:  Night → Reduced Natural Light
Answer:  At night, the lack of natural light darkens the park, with only artificial
lights visible.
```

---

---

Condition 1 Prompt (continue)

*Question:* How would the scene differ if a month-long drought had preceded this day?
```
Causal Chain:  Drought → Dry Soil → Wilted Vegetation → Sparse Leaves
Answer:  The drought dries the soil, leading to wilted, brown grass and sparse tree
leaves.
```
**Provided Information:**
Question: {}
**Task:** Answer the provided question based on the image, optionally providing a lightweight causal chain to represent the causal reasoning followed by a text response, or providing only a text response that explains the causal reasoning. Follow the instructions above and output in the appropriate format.

---

### A.1.2. CONDITION 2: PARTIAL SCAFFOLDING

This condition removes the three worked examples but retains all detailed instructions, Pearl framework references, and format templates.

---

Condition 2 Prompt

**System:** You are a vision-language model (VLM) tasked with answering a provided question about an image in a way that demonstrates causal reasoning, based on Judea Pearl's framework. You are encouraged to construct a lightweight causal chain to represent the chain of cause-and-effect relationships relevant to the answer, using the number of nodes and arrows needed to accurately capture the causal logic (e.g., 2 nodes like A → B, 3 nodes like A → B → C, or more if appropriate), tailored to the complexity of the reasoning. However, it is reasonable to provide only a text response without a causal chain if you find it sufficient to explain the causal reasoning. If you include a chain, provide it first, followed by a concise text response that explains the reasoning based on the chain. If you omit the chain, provide only the text response, ensuring it still reflects causal reasoning.
**Instructions:**
- **Base the response on the image and question:** Ensure the answer (and chain, if provided) is grounded in the image's content and relevant to the question.
- **Optional causal chain:** If you choose to include a chain, follow these steps:
    1. Identify the key cause-and-effect relationships implied by the question (e.g., 'Rain causes wet ground, which darkens the scene').
    2. Format these relationships as a chain using '→' to connect nodes (e.g., Rain → Wet Ground → Darkened Scene).
    3. Use the fewest nodes needed for clarity, but include additional nodes if the causal chain requires multiple steps (e.g., 2, 3, or more nodes).
- **Provide a concise answer:** Write 1–2 sentences that explain the causal reasoning, referencing the chain's steps if a chain is provided, or directly addressing the question's causal logic if no chain is included.
- **Avoid overly simplistic chains:** Ensure the response (and chain, if provided) captures specific, relevant causal relationships (e.g., avoid vague terms like 'Change → Outcome' unless fully justified).
- **Format the output exactly as follows:**

    If including a chain:
    ```
    Causal Chain:  [Your causal chain, e.g., A → B or A → B → C → D]
    Answer:  [Your answer, explaining the reasoning based on the chain]
    ```
    If omitting the chain:
    ```
    Answer:  [Your answer, explaining the causal reasoning]
    ```
**Provided Information:**
Question: {}
**Task:** Answer the provided question based on the image, optionally providing a lightweight causal chain to represent the causal reasoning followed by a text response, or providing only a text response that explains the causal reasoning. Follow the instructions above and output in the appropriate format.

---

### A.1.3. CONDITION 3: MINIMAL SCAFFOLDING

This condition removes both examples and detailed instructions, providing only a basic task description with Pearl framework reference and minimal node count examples.

---

**Condition 3 Prompt**

**System:** You are a vision-language model (VLM) tasked with answering a provided question about an image in a way that demonstrates causal reasoning, based on Judea Pearl's framework. You are encouraged to construct a lightweight causal chain to represent the chain of cause-and-effect relationships relevant to the answer, using the number of nodes and arrows needed to accurately capture the causal logic (e.g., 2 nodes like Rain → Wet Ground, 3 nodes like Rain → Wet Ground → Fewer People, or more if appropriate), tailored to the complexity of the reasoning. However, it is reasonable to provide only a text response without a causal chain if you find it sufficient to explain the causal reasoning. If you include a chain, provide it first, followed by a concise text response that explains the reasoning based on the chain. If you omit the chain, provide only the text response, ensuring it still reflects causal reasoning.
**Provided Information:**
Question: {}
**Task:** Answer the provided question based on the image, optionally providing a lightweight causal chain to represent the causal reasoning followed by a text response, or providing only a text response that explains the causal reasoning. Follow the instructions above and output in the appropriate format:
If including a chain:
```
Causal Chain:  [Your causal chain]
Answer:  [Your answer]
```
If omitting the chain:
```
Answer:  [Your answer]
```

---

### A.1.4. KEY DIFFERENCES ACROSS CONDITIONS

Table 4 summarizes which scaffolding components are present in each condition.

*Table 4.* Scaffolding Components Across Instruction Conditions

| Component | Condition 1 | Condition 2 | Condition 3 |
|---|---|---|---|
| Worked Examples (3) | ✓ | × | × |
| Detailed Instructions | ✓ | ✓ | × |
| Step-by-Step Guidance | ✓ | ✓ | × |
| Format Templates | ✓ | ✓ | ✓ |
| Pearl Framework Reference | ✓ | ✓ | ✓ |
| Node Count Examples | ✓ | ✓ | ✓ |

This progression isolates the effects of worked examples (Condition 1 vs. 2) and detailed instructions (Condition 2 vs. 3) while holding constant Pearl framework references and basic format guidance. Models with robust internal causal representations should maintain performance across conditions; degradation indicates dependence on specific scaffolding components.

### A.2. Detailed Evaluation and Scoring Methodology

We implement a rigorous, granular scoring system for evaluating VLM responses. This system is designed to assess different facets of visual causal reasoning performance, from textual plausibility to the generation of explicit causal structures. The evaluation is guided by a dedicated prompt provided to both automated judges (GPT-4o and Claude 4.5 Sonnet) and human annotators, with strict instructions on scoring criteria and the required output format.

Key aspects of our scoring methodology include:

- **Scoring Scale:** All responses are scored on a 0-10 scale. A score of 0 indicates a missing or entirely irrelevant response. Higher scores reflect increasing accuracy, relevance, logical consistency, and adherence to level-specific requirements, with 10 representing a fully correct, precise, and well-supported answer.

- **Text Response Criteria:** Text responses are evaluated on multiple dimensions:
  - *Accuracy:* Factual correctness of statements relative to the provided image and captions.
  - *Relevance:* How directly and completely the response addresses the question asked.
  - *Logical Consistency:* Adherence to common-sense causal reasoning principles and the plausibility of the causal links implied in the text.

- *Level-Specific Requirements:* Fulfillment of criteria specific to the causal level of the question (L1, L2, L3), such as identifying patterns (L1) or describing plausible outcomes/scenarios (L2/L3). Specific deduction rules are applied for errors appropriate to each level (e.g., factual errors for L1, insufficient causal steps or missing temporal/environmental shifts for L2/L3).

- **Causal chain Criteria (L2 & L3):** For Level 2 and 3 questions, the explicitly generated causal chains are evaluated based on:

  - *Structural Validity:* The output adheres to a proper chain format with arrows connecting causal elements.
  - *Minimum Link Requirements:* The chain includes at least one causal link ($\geq 1$) for Level 2 questions and at least two causal links ($\geq 2$) for Level 3 questions, reflecting the minimum expected complexity for these causal levels.
  - *Causal Coherence:* The overall chain structure represents a coherent, logical, and plausible causal pathway relevant to the question.

- **Level 2/3 Specific Scoring Protocol:** Each Level 2 and 3 response receives two separate scores: one specifically for the generated causal chain based on the criteria above, and another for the text response. This allows us to differentiate between a model's ability to generate structured causal knowledge and its ability to articulate it linguistically.

- **Chain-Text Consistency:** In addition to scoring text and chain independently, evaluators assess the consistency and alignment between the generated chain and the accompanying text explanation. Significant misalignments (e.g., chain contradicting the text, text explaining links not present in the chain) indicate a potential disconnect between structured and linguistic representations, suggesting superficial rather than deeply integrated causal understanding.

- **Strict Output Format:** Judges are strictly instructed to output only the numerical scores in a specific machine-readable format (e.g., '[0][8][6][7]...'), without any additional explanatory text. This ensures structured outputs amenable to quantitative analysis.

The comprehensive scoring rubrics, detailed criteria, specific deduction rules, and the exact evaluation prompt provided to the automated judges and human annotators are included in Appendix A.3. This rigorous system, combined with the use of multiple independent judges and human validation, significantly enhances the reliability and robustness of our evaluation process.

### A.3. Evaluation Prompt

We present the evaluation prompt used to instruct the automated judges, GPT-4o and Claude 4.5 Sonnet, to assess the quality of both the textual responses and the causal chains generated by the candidate VLMs. The judges are asked to evaluate text and chain responses using 0-10 scales. Text evaluates accuracy, relevance, and logical consistency. Chains additionally assess structural validity (correct arrow notation), link requirements, and causal coherence. The placeholders {} indicate where the specific image's captions and VLM's responses are inserted.

---

**Evaluation Prompt**

**System:** You are a vision-language model (VLM) evaluating a candidate VLM's responses to a visual causal reasoning task based on Judea Pearl's Ladder of Causation: Level 1 (Association) identifies observable scene patterns; Level 2 (Intervention) predicts outcomes of a specific change, considering temporal/environmental factors; Level 3 (Counterfactual) reasons about alternate scenarios if a past condition differed, requiring multi-step causal logic.
You will be provided with an image, 5 captions describing the image, 9 questions about the image (3 per Ladder level, with Level 3 questions emphasizing abstract reasoning), and the candidate VLM's 9 responses. For Levels 2 and 3, each response is expected to include a lightweight causal chain (e.g., A $\rightarrow$ B or A $\rightarrow$ B $\rightarrow$ C) to explain the reasoning.
Your task is to evaluate each response by assigning two separate scores from 0 to 10: one for the causal chain (if applicable) and one for the text response, using the image, captions, common-sense reasoning, and, for Levels 2 and 3, the causal chain to judge accuracy, relevance, and logical consistency. No ground truth answers are provided, but you must rely on your understanding of the scene and causal principles to make informed judgments.
**Instructions for Evaluation**
**Scoring Criteria:** Assign a score from 0 to 10 based on how well the candidate VLM's response answers the question, considering its accuracy, relevance, and logical consistency with the image, captions, and common-sense reasoning. For Levels 2 and 3, also evaluate the causal chain and its alignment with the text.

---

---

Evaluation Prompt (continue)

- **Causal chain (Levels 2 and 3 only):**
  - (8–10): Fully correct, relevant, with required links (Level 2: $\geq 1$, e.g., `Rain → Wet Path`; Level 3: $\geq 2$, e.g., `Midnight → Dark Sky → Stars`).
  - (6–7): Partially correct, misses minor links or has small inaccuracies (e.g., Level 3: `Midnight → Stars`, omitting `Dark Sky`).
  - (3–5): Incorrect, with illogical or irrelevant links (e.g., `Rain → Sun`).
  - (1–2): Provided but severely flawed (e.g., wrong direction, no causal logic).
  - (0): Missing or not provided.

- **Text Response (All Levels):**
  - (8–10): Fully correct, addresses question precisely, aligns with chain for Levels 2 and 3 (e.g., Level 3: "Stars visible" with `Midnight → Dark Sky → Stars`).
  - (6–7): Partially correct, addresses question but misses details or minor causal steps (e.g., Level 2: "Path wet" without slipperiness).
  - (3–5): Incorrect, misinterprets question or contradicts image/captions (e.g., Level 1: "Grass is blue" when green).
  - (1–2): Provided but severely flawed (e.g., irrelevant to question).
  - (0): Missing or not provided.

**Deductions:**

- Level 1: Deduct 1–3 points for factual errors (e.g., wrong color, position).

- Level 2: Deduct 2–4 points if chain lacks $\geq 1$ link or text omits environmental/temporal shifts (e.g., no mention of rain's effect).

- Level 3: Deduct 3–5 points if chain lacks $\geq 2$ links (e.g., `Snow → Ground White`, missing `Cold → Icicles`) or text omits multi-step reasoning/shifts.

- All Levels: Deduct 1–2 points for overgeneralization (e.g., vague "things change") or ignoring image/captions.

**Use the Image and Captions:** Ensure responses are grounded in the image and captions for Level 1 questions. For Levels 2 and 3, expect reasonable speculation about changes (e.g., weather, time) that align with common-sense dynamics.
**Evaluation Guidelines:**

- Level 1 (Association): Score text for accuracy against image/captions (e.g., "What color is the grass?" → "Green" if green). No chains are expected. Set chain score to [0].

- Level 2 (Intervention): Score text for plausible outcome (e.g., "If rain starts, path becomes wet") and chain for $\geq 1$ logical link (e.g., `Rain → Wet Path`). Ensure text and chain align (e.g., text mentions wetness, chain includes it). Check for temporal/environmental shifts (e.g., weather change).

- Level 3 (Counterfactual): Score text for plausible alternate scenario (e.g., "If midnight, stars visible") and chain for $\geq 2$ logical links (e.g., `Midnight → Dark Sky → Stars`). Ensure text and chain align and reflect multi-step reasoning (e.g., temporal shift to night). Deduct heavily for missing shifts or single-link chains.

**Output Format Instructions** Output your evaluation in exactly this format, with no additional text, line breaks, spaces, explanations, or deviations:

- `[Chain Rating for Response 1][Text Rating for Response 1][Chain Rating for Response 2][Text Rating for Response 2][Chain Rating for Response 3][Text Rating for Response 3][Chain Rating for Response 4][Text Rating for Response 4][Chain Rating for Response 5][Text Rating for Response 5][Chain Rating for Response 6][Text Rating for Response 6][Chain Rating for Response 7][Text Rating for Response 7][Chain Rating for Response 8][Text Rating for Response 8][Chain Rating for Response 9][Text Rating for Response 9]`

- Each `[Chain Rating]` and `[Text Rating]` is a number from 0 to 10 in square brackets (e.g., `[7]`).

- For Level 1, use `[0]` for the chain score.

- Do not include explanations, extra spaces, line breaks, headings, or any text outside this format.

---

**Evaluation Prompt (continue)**

Example Output: `[0][8][0][7][0][9][6][8][8] [9][7][6][9][9][5][5][8][8]`
**Provided Information** 5 Captions: `{}`
`9 pairs of Questions and Candidate VLM's Responses: {}`
**Task** Evaluate the candidate VLM's responses to the 9 questions using the provided image, captions, and instructions above. Assign separate scores for the causal chain and text response for each, and output your evaluation in the following format and nothing else:
**Output Format** `[Chain Rating for Response 1][Text Rating for Response 1][Chain Rating for Response 2][Text Rating for Response 2][Chain Rating for Response 3][Text Rating for Response 3][Chain Rating for Response 4][Text Rating for Response 4][Chain Rating for Response 5][Text Rating for Response 5][Chain Rating for Response 6][Text Rating for Response 6][Chain Rating for Response 7][Text Rating for Response 7][Chain Rating for Response 8][Text Rating for Response 8][Chain Rating for Response 9][Text Rating for Response 9]`

## A.4. Experimental Results with Confidence Intervals

Tables 5 and 6 provide the results of the experiments conducted in Section 3 of the main paper with the 95% confidence intervals recorded. 95% confidence intervals' widths are relatively narrow, with a width range from 0.1 to 0.3 for most scores.

*Table 5.* Level 1 (Association) Scores Across Instruction Conditions with 95% Confidence Intervals. Text scores remain stable, indicating preserved basic visual understanding. Scores averaged over 500 images, 3 questions.

| VLM | GPT-4o | | | Claude 4.5 Sonnet | | |
|---|---|---|---|---|---|---|
| | Cond 1 | Cond 2 | Cond 3 | Cond 1 | Cond 2 | Cond 3 |
| LLaVA-NeXT | 8.04 (7.94-8.15) | 7.99 (7.89-8.09) | 7.97 (7.88-8.08) | 7.37 (7.29-7.46) | 7.77 (7.70-7.83) | 7.68 (7.61-7.75) |
| LLaVA-CoT | 8.06 (7.96-8.16) | 7.98 (7.89-8.10) | 7.97 (7.87-8.08) | 7.86 (7.79-7.94) | 7.84 (7.77-7.92) | 7.80 (7.71-7.87) |
| LLaVA-RLHF | 5.06 (4.92-5.20) | 5.24 (5.16-5.42) | 5.17 (5.07-5.33) | 5.15 (5.04-5.30) | 5.13 (5.05-5.30) | 5.04 (4.95-5.21) |

*Table 6.* Levels 2-3 (Intervention & Counterfactual) Scores with 95% Confidence Intervals. Format: (Text, **Chain**). The last two subrows of each model are the 95% confidence intervals of the text and chain generated by that model, respectively. Chain degradation substantial as scaffolding removed. Averaged over 500 images, 3 questions/level. Blanks scored 0. [†]LLaVA-RLHF: 2,467 blanks (Cond 1), 18 (Cond 2).

| VLM | GPT-4o | | | | | | Claude 4.5 Sonnet | | | | | |
|---|---|---|---|---|---|---|---|---|---|---|---|---|
| | Cond 1 | | Cond 2 | | Cond 3 | | Cond 1 | | Cond 2 | | Cond 3 | |
| | L2 | L3 | L2 | L3 | L2 | L3 | L2 | L3 | L2 | L3 | L2 | L3 |
| LLaVA-NeXT | (8.25, **7.95**) | (7.85, **7.42**) | (7.79, **7.09**) | (7.42, **6.31**) | (7.58, **6.15**) | (7.25, **5.59**) | (7.66, **7.70**) | (7.42, **7.79**) | (6.98, **7.11**) | (6.71, **6.75**) | (6.92, **6.48**) | (6.57, **5.95**) |
| | (8.19-8.31) | (7.79-7.92) | (7.72-7.86) | (7.36-7.49) | (7.49-7.64) | (7.17-7.30) | (7.62-7.70) | (7.38-7.48) | (6.92-7.03) | (6.68-6.79) | (6.86-7.00) | (6.49-6.65) |
| | **(7.87-8.02)** | **(7.35-7.51)** | **(6.99-7.18)** | **(6.21-6.40)** | **(6.04-6.29)** | **(5.49-5.71)** | **(7.66-7.74)** | **(7.75-7.84)** | **(7.05-7.18)** | **(6.69-6.83)** | **(6.37-6.58)** | **(5.85-6.06)** |
| LLaVA-CoT | (7.26, **0.77**) | (7.16, **1.34**) | (7.35, **0.73**) | (7.13, **0.97**) | (7.45, **2.28**) | (7.19, **2.41**) | (6.30, **1.17**) | (6.45, **1.86**) | (6.55, **1.04**) | (6.10, **0.89**) | (6.67, **2.46**) | (6.13, **2.17**) |
| | (7.21-7.32) | (7.10-7.23) | (7.30-7.42) | (7.08-7.21) | (7.37-7.54) | (7.09-7.28) | (6.26-6.37) | (6.40-6.52) | (6.50-6.60) | (6.06-6.17) | (6.60-6.74) | (6.04-6.21) |
| | **(0.66-0.90)** | **(1.22-1.51)** | **(0.64-0.87)** | **(0.86-1.11)** | **(2.12-2.47)** | **(2.30-2.64)** | **(1.04-1.32)** | **(1.70-2.04)** | **(0.90-1.18)** | **(0.76-1.01)** | **(2.26-2.62)** | **(2.02-2.35)** |
| LLaVA-RLHF[†] | (1.23, **0.51**) | (1.92, **1.11**) | (6.38, **0.70**) | (6.39, **0.47**) | (6.52, **0.79**) | (6.73, **0.92**) | (1.43, **1.11**) | (1.85, **1.52**) | (5.51, **0.99**) | (5.34, **0.70**) | (5.59, **0.88**) | (5.60, **0.69**) |
| | (1.10-1.37) | (1.77-2.10) | (6.32-6.47) | (6.37-6.51) | (6.46-6.61) | (6.69-6.83) | (1.28-1.57) | (1.68-1.99) | (5.47-5.59) | (5.30-5.42) | (5.55-5.67) | (5.55-5.68) |
| | **(0.41-0.60)** | **(0.98-1.24)** | **(0.59-0.81)** | **(0.39-0.56)** | **(0.69-0.90)** | **(0.85-1.06)** | **(0.97-1.24)** | **(1.34-1.65)** | **(0.86-1.10)** | **(0.59-0.80)** | **(0.77-1.00)** | **(0.59-0.80)** |

