# OpenReview forum: "Position: What Cézanne Knew About Visual Intelligence That Vision-Language Models Miss"
_ICML.cc/2026/Position_Paper_Track — ICML 2026 Position Paper Track spotlight_

### Official Review · Reviewer_XRVD · 2026-02-15

**Significance:** 3
**Argument Clarity:** 3
**Rating:** 5
**Confidence:** 3

**Questions:**

1. The discussions on temporal constitution might need further clarification. Even though I do not fully agree with the scope of the position, I think it is a good position paper for discussions. I am willing to change my ratings accordingly if the discussions are meaningful.

**Alternative Views Section:**

Yes

**Compliance With Llm Reviewing Policy A Conservative:**

Affirmed.

**Discussion Potential:**

4

**Paper Summary:**

The paper claims that current VLM benchmarks for causal reasoning have two major blind spots: 1. Presuppose temporal constitution without testing whether it is a prerequisite. 2. Insufficiency in distinguishing external symbolic scaffolding from internalized capability. In this view, the authors work on the frameworks of art, philosophy, and psychoanalysis and show absent constitutive internalization.

**Position:**

Yes

**Position In Title:**

Yes

**Related Work:**

3

**Strengths And Weaknesses:**

Strength:
1. The discussions on the plausibility-faithfulness gap are inspiring, calling for future research for further exploration.
2. The alternative views and discussions on that are meaningful, solving some of my concerns about this paper (especially on the discussions of scaling law and video model).
3. From my perspective, this is a good discussion, highlighting a potential path that the current community might have ignored.

Weakness:
1. Would the claim "causal reasoning is inherently temporal" overclaim? As stated in [1-3], it seems that counterfactuals can be reached via token-level editing, which does not align with the claim that "alternative pasts would produce different presents". But more focus on single-frame alternations. This is a fundamental question as the paper is built on the temporal constitution. More discussions should be mentioned here.

2. Also, the example given (C´ezanne) is not very intuitive. Why is observing from different positions understood as multiple moments? If the idea of "over time" is not essential (e.g., can use multiple camera positions to capture the object simultaneously), then, similar to this claim, we can argue that spatial knowledge is more important (we can thus focus on spatial accuracy instead of temporal constitution).

3. It is really good to see that several models are evaluated, and none of the tested models inhabit the causal reasoning framework as internal structure. It might be good (but not required) to train a small model under the authors' current assumption to validate the future of this position. Also, current evaluation on "relatively" small models might be biased, as they may not reveal the full capacity of understanding the prompt instruction (mentioned in Sec. 5, but my claim here is different: not because of the flexibility, but due to the limited instruction following ability provided by the small models). At least one larger model is expected.

[1] Reft: Representation finetuning for language models.

[2] Re-Imagining Multimodal Instruction Tuning: A Representation View

[3] A concept-based explainability framework for large multimodal models

**Support:**

3

---

> ### Author Rebuttal · Authors · 2026-03-30
>
> We thank the reviewer for the engaged critique.
>
> **On temporal constitution and token-level editing.** This is an important challenge: if counterfactual outputs can be produced via representation editing (ReFT, [1-3]) without temporal understanding, does our position overreach?
>
> We clarify what we mean by temporal constitution. Temporal constitution is the capacity to represent time as the medium through which causes produce effects: encoding that causes precede effects, that causal processes unfold with duration, and that present states constrain future possibilities. This differs from temporal inference (e.g., seeing rust and outputting "old"), which extracts associations from static markers without representing the oxidation process that connects past to present.
>
> A concrete example: consider an image of a cracked sidewalk next to a tree. A model performing temporal inference might output "the tree roots caused the crack" by associating spatial proximity with causation. A model with temporal constitution would represent the process: roots grow over years, exert gradual pressure on the concrete, stress accumulates until the fracture threshold, crack propagates. The first produces correct output for familiar scenes; the second generalizes to novel causal scenarios because it encodes the underlying temporal-mechanical process.
>
> We also clarify the scope. We do not argue that *all counterfactual operations* require temporal constitution. Token-level interventions can produce counterfactual outputs by editing representations, bypassing whatever internal structure the model possesses. Our claim concerns *genuine causal reasoning about depicted scenes*: the capacity to model how causes propagate through time to produce effects, supporting generalization to novel scenarios.
>
> The ReFT literature supports our diagnostic approach. Representation interventions succeed because the *external agent* imposes counterfactual structure on the model's representations. The model does not generate this structure internally. This parallels our scaffolding analysis: when external support (scaffolding or representation editing) produces correct outputs, we cannot conclude the model has internalized causal reasoning.
>
> **On Cézanne and space vs. time.** The reviewer observes that multiple viewing positions could be captured simultaneously, making "over time" inessential. The core contrast is not spatial vs. temporal, but *extracted* vs. *constructed*:
>
> - **Extracted:** A camera captures features from a frozen moment.
> - **Constructed:** Understanding is built through active engagement, integrating information and updating representations.
>
> Cézanne built his representation through sustained perceptual engagement, not by setting up four cameras simultaneously. For VLMs, the question is whether outputs reflect extracted patterns or constructed understanding. Our diagnostics provide evidence: scaffolding invariance suggests constructed understanding; scaffolding dependence suggests pattern extraction relying on external support.
>
> The position paper aims to stimulate discussion toward new algorithmic approaches grounded in constructive processing. Benchmarks that systematically vary scaffolding can expose the gap between current capabilities and genuine causal construction, guiding future VLM development. We acknowledge the reviewer's point that spatial constitution may also matter; temporal constitution is one prerequisite we identify, not the only possible one.
>
> **On model scale and instruction-following capacity.** We have tested models spanning 8B to 76B parameters across four structural output formats: three linearized chain notations (arrow, semicolon, numbered) and one graph format (adjacency list). Two findings address the concern.
>
> Qwen3-VL-8B-Instruct (2025) achieves the highest performance among all tested models despite being the smallest. For linearized formats, Qwen3-VL achieves CDD near zero across all scaffolding conditions. If limited instruction-following capacity explained the gap, larger models would outperform smaller ones; they do not.
>
> InternVL2 76B (the largest model tested) still exhibits a substantial plausibility-faithfulness gap. On counterfactual questions requiring causal chain generation, InternVL2 achieves text quality of 7.46 but chain quality of only 2.42. Strong instruction-following capacity at 76B does not eliminate this disparity.
>
> Even Qwen3-VL exhibits scaffolding dependence for structurally complex outputs:
>
> | Condition | Compliance | CDD |
> |:----------|:----------:|:---:|
> | Full scaffolding (C1) | 98% | 0.02 |
> | Minimal scaffolding (C3) | 44% | 0.22 |
>
> Standard evaluation under fixed scaffolding would conclude Qwen3-VL possesses robust causal reasoning. Varying scaffolding shows the capability is partially scaffolding-dependent when structural demands increase. Both the smallest and largest models tested exhibit patterns our diagnostics detect; scale does not determine scaffolding sensitivity.

---

> > ### Author Rebuttal · Reviewer_XRVD · 2026-03-31
> >
> > Thank you for the excellent rebuttal, which has solved all my concerns. It would be great if the authors could incorporate these discussions in the final revision, which can strengthen the claim of the position.

---

### Official Review · Reviewer_mNPd · 2026-03-12

**Significance:** 3
**Argument Clarity:** 1
**Rating:** 4
**Confidence:** 3

**Questions:**

Could the authors make it more clear about what is the relation and analogy of Figure 1 and the stated position? Also, what are some more concrete examples to share with readers on the blind spots, the hypothesis, and evaluation process proposed in the paper?

**Alternative Views Section:**

Yes

**Compliance With Llm Reviewing Policy A Conservative:**

Affirmed.

**Discussion Potential:**

3

**Final Justification:**

This position paper presents a valid concern on whether VLMs truly understand the causal consequences, which is worthwhile for the community to look into and discuss. Additional results on other VLMs beyond LLaVA are presented in the rebuttal to further enrich the supporting evidence. The only concern that remains is the clarity of the paper writing, as the current version is too abstract and vague, with a somewhat improper analogy that makes the understanding of the paper even harder. I think the paper would need a revision on the clarity of the text to include more concrete examples, along with detailed analysis of the current fail modes of the existing VLMs. I would keep the rating at borderline accept.

**Paper Summary:**

This paper presents a position that we should pay effort to check whether the current VLMs can understand that the causal consequences come from the medium of time. The paper proposes two blind spots in the causal reasoning capability evaluation in VLMs, which are the temporal prerequisite and the symbolic diagnostic. Preliminary experiments demonstrate that VLMs still struggle to yield valid causal structure, so the paper also calls benchmark developers and VLM researchers to pay attention to the validity of causal structure in VLM reasoning.

**Position:**

Yes

**Position In Title:**

Yes

**Related Work:**

2

**Strengths And Weaknesses:**

**Strengths:**

- The paper presents a position that is often overlooked in VLM evaluation, which states that we should examine whether VLMs really have the correct causal chain structure during reasoning, instead of simply presuming that VLMs already know about temporal constitution. Therefore, it can inspire in-depth discussions and debates on whether we should focus on this issue, and how we are going to properly address and evaluate it.

- The stated position is supported by the preliminary experimental analysis on the LLaVA family. There is evidence supporting both hypothesis for the current reasoning status of VLMs, followed by analysis on the results to explain how scaffolding functions in different VLMs.

- The calls to action and suggestions for benchmark developers and VLM researchers are detailed and insightful, which can provide guidance for readers to see which directions to go for to make progress on causal reasoning of VLMs.


**Weaknesses:**

- The presentation is sometimes to obscure and hard to understand. For example, the explanation on the two identified blind spots in Section 2 uses too many abstract descriptions such as the theory or quotes from psychology or cognitive science (e.g., right column on Lines 103-104, left column on Lines 130-134, left column on Lines 144-145, right column on Lines 126-128). For people who are not familiar with these interdisciplinary references (I think I am not alone in the machine learning community to be unfamiliar with these psychology or cognitive science stuff), it adds a lot of burdens to read through and understand the paper. What makes me feel even more confusing is the analogy to the Cezanne's work in Lines 110-128. Actually, I cannot understand what is the internal relationship between Cezanne's geometric analysis (in Lines 110-128 and Figure 1) and the stated position of this paper. Analogies are used to make readers easier to understand the concept, not to make it more complicated. All these obscure makes this paper difficult to read and understand. It will be better if the paper can refer to more concrete examples when explaining the blind spots, the hypothesis, and evaluation process.

- The experimental analysis is only restricted to the LLaVA family. Although LLaVA being the pioneering model in VLMs, I think the Qwen and InternVL series may be the more popular open-source VLMs these days, which are also worthwhile to study and analysis. Also, I think the preliminary experiments also do not have to restrict to open-source models, because it does not evolve model finetuning. Therefore, more popular models like GPT-4o and its follow-ups, the Gemini series are also worthwhile to investigate on.

**Support:**

3

---

> ### Author Rebuttal · Authors · 2026-03-30
>
> We thank the reviewer for the feedback on accessibility.
>
> **Question 1: What is the relation between Fig 1 and the stated position?**
>
> A camera captures a single instant from a single position. Cézanne painted differently. In Figure 1b, the geometric analysis shows four labeled positions marking different eye levels, with each object rendered from a separate vantage point. This is not how a photograph works. Cézanne's multi-viewpoint technique exposes how humans encode temporal experience into a coherent representation: understanding built through sustained engagement, not extracted from a frozen moment.
>
> We use this as a framework for what genuine causal understanding requires. An AI capable of true causal reasoning should encode time analogously: representing that causes precede effects, that processes unfold with duration, and that present states constrain future possibilities.
>
> The connection to VLMs: consider a model shown an image of a rusted bicycle and asked why it looks this way. It outputs: "This bicycle is old because of the rust." Two very different processes could produce this answer:
>
> - **Camera-like processing:** The model pattern-matches "rust → old" from training data, extracting a statistical association from the frozen image.
> - **Cézanne-like processing:** The model understands that rust accumulates through oxidation over time, that this process is irreversible, and that the bicycle's current state constrains its possible futures.
>
> Both produce the same output, but only the second reflects genuine temporal understanding. Current benchmarks measure whether the output is correct; they cannot distinguish these underlying processes. Our diagnostics (CDD and SS) make this distinction testable.
>
>
> **Question 2: Concrete examples for the blind spots, hypothesis, and evaluation process.**
>
> *Temporal blind spot and H1:*
> - **What we test:** Given an image, we ask the model to produce (a) a text explanation and (b) a causal chain using arrow notation (e.g., "strong wind → force on umbrella → umbrella tips over").
> - **What we measure:** Causal Depth Disparity (CDD) = the gap between text quality and chain validity, normalized by text quality.
> - **What results mean:** A model might write a fluent paragraph explaining why umbrellas tip in the wind, but fail to produce a valid three-step chain. A high CDD (near 1.0) indicates that the model pattern-matches causal language without constructing structured causal representations. Low CDD (near 0) indicates text and structure are aligned.
>
> *Symbolic blind spot & H2:*
> - **What we test:** We present identical questions under three scaffolding conditions:
>   - C1 (Full): Detailed instructions, Pearl's causal framework, worked examples, output templates
>   - C2 (Partial): Instructions and framework, no worked examples
>   - C3 (Minimal): Brief task description only
> - **What we measure:** Scaffolding Sensitivity (SS) = chain quality at C1 minus chain quality at C3.
> - **What results mean:** Positive SS indicates the model depends on scaffolding; negative SS indicates scaffolding interferes. Divergent responses across models (LLaVA-NeXT: SS = +1.83; LLaVA-CoT: SS = −1.07) indicate that each model relates to causal structure differently based on training history. None inhabit the causal framework as internal structure.
>
> **On model selection.** Beyond 3 LLaVA models, we conducted additional experiments testing 6 VLMs across four structural formats and three scaffolding levels. The additional models include CogVLM2 (2024), MiniGPT-4 (2023), and Qwen3-VL-8B-Instruct (2025). The four formats include three linearized chain notations (arrow, semicolon, numbered) and one graph format (adjacency list).
>
> | Model | Numbered (C1) | Arrow (C1) | Semicolon (C3) | Adj. List (C3) |
> |:------|:-------------:|:----------:|:--------------:|:--------------:|
> | LLaVA-NeXT | −0.01 | 0.04 | 0.36 | 0.70 |
> | LLaVA-RLHF | −0.05 | 0.01 | 0.48 | 0.86 |
> | LLaVA-CoT | 0.00 | 0.06 | 0.06 | 0.22 |
> | CogVLM2 | 0.33 | 0.38 | 0.68 | 0.82 |
> | MiniGPT-4 | 0.00 | 0.13 | 0.17 | 0.57 |
> | Qwen3-VL (2025) | −0.01 | 0.01 | 0.01 | 0.22 |
>
> *Table 1: Causal Depth Disparity (CDD) across formats. Values near zero indicate text-structure alignment; high values indicate the plausibility-faithfulness gap.*
>
> Qwen3-VL (2025) achieves the best performance among tested models for linearized formats (CDD near 0). Standard evaluation under fixed scaffolding would conclude this model has robust causal reasoning. Varying scaffolding shows otherwise: for adjacency lists, format compliance drops from 98% (C1) to 44% (C3). Even the strongest model exhibits scaffolding dependence when structural demands increase. The patterns hold beyond the LLaVA family.
>
> **On improving accessibility.** We will revise Sec 2 to lead with concrete VLM examples (e.g., a rusted bicycle) before introducing theoretical frameworks. The interdisciplinary references will be positioned as a formalization of intuitions already established through examples.

---

> > ### Author Rebuttal · Reviewer_mNPd · 2026-04-02
> >
> > Dear authors,
> >
> > Thanks for your rebuttal! For the analogy between Cézanne's painting and the stated position, I still think the analogy is not helpful for people to understand the paper. A good analogy should provide audience with easier understanding to the paper, instead of making it more complicated and requiring more efforts to think about it. This is not a feeling from only myself, as I also see Reviewer XRVD has this comment about the example of Cézanne's painting is not very intuitive. I think the paper needs revision on the text to add much more concrete examples on what problems the paper is solving (like what the authors have replied for Question 2), for the paper to be easier to understand for audience. Otherwise, the current version of the paper is too obscure and abstract.
> >
> > Apart from this, I feel the rest of my concerns are addressed.

---

### Official Review · Reviewer_P3Kr · 2026-03-13

**Significance:** 2
**Argument Clarity:** 2
**Rating:** 4
**Confidence:** 3

**Questions:**

The questions are about the weaknesses. Please give explanations about the weaknesses mentioned before.

**Alternative Views Section:**

Yes

**Compliance With Llm Reviewing Policy A Conservative:**

Affirmed.

**Discussion Potential:**

3

**Final Justification:**

The authors addresses my concerns in the rebuttal. I raise my rating.

**Paper Summary:**

As vision-language models (VLMs) are increasingly deployed in high-stakes domains like medicine and autonomous driving, accurately evaluating their causal reasoning capabilities has become critical. This position paper argues that current VLM causal reasoning benchmarks suffer from two major blind spots: they presuppose an inherent understanding of time without testing it, and they fail to distinguish genuine internalized knowledge from reliance on external prompt scaffolding. Consequently, existing evaluations often mask a dangerous gap between plausible-sounding, fluent linguistic outputs and truly faithful, structured causal reasoning. Preliminary evidence demonstrates that when external scaffolding is altered or removed, VLM performance degrades significantly, revealing a lack of true constitutive understanding. To address these vulnerabilities, the authors urge the research community to develop new benchmarks that rigorously test foundational temporal understanding and scaffolding-invariance rather than merely measuring superficial output accuracy.

**Position:**

Yes

**Position In Title:**

Yes

**Related Work:**

2

**Strengths And Weaknesses:**

### Strengths
1. The paper introduces a compelling theoretical framework by identifying temporal constitution and scaffolding-invariance as fundamental, previously overlooked prerequisites for genuine causal reasoning in vision-language models.

2.  This paper diagnoses the dangerous plausibility-faithfulness gap, effectively explaining why current models can generate fluent, convincing narratives that actually lack valid underlying causal structures.

3. By drawing on interdisciplinary concepts from art, philosophy, and psychoanalysis, the authors propose a highly original diagnostic methodology that moves beyond superficial accuracy metrics to deeply probe internalized model capabilities.

### Weaknesses
1. The evaluation requires models to generate causal chains using a specific arrow notation. Poor performance, such as catastrophic blank responses, likely stems from simple formatting conflicts or failures in instruction following rather than a genuine lack of underlying causal reasoning.

2. The authors draw sweeping conclusions about the fundamental limitations of current VLMs while only testing older open-source models from the LLaVA family, without evaluating state-of-the-art, (e.g., Qwen3-VL, InternVL3.5, Gemini 3, etc.).

3. The study uses single-frame, static images from the COCO dataset to test Level-3 counterfactual reasoning. This contradicts the paper's own theoretical emphasis on "temporal constitution" as the medium for causality.

**Support:**

2

---

> ### Author Rebuttal · Authors · 2026-03-30
>
> We thank the reviewer for engaging with the methodological concerns.
>
> **Additional experiments.** We have conducted experiments beyond the position paper, testing six VLMs across four structural output formats and three scaffolding levels. The six models span multiple architecture families: LLaVA-NeXT (2024), LLaVA-RLHF (2024), LLaVA-CoT (2025), CogVLM2 (2024), MiniGPT-4 (2023), and Qwen3-VL-8B-Instruct (2025). The four formats include three linearized chain representations and one graph representation:
>
> - **Linearized formats:** Arrow (A → B → C), Semicolon (A; B; C), Numbered (1. A  2. B  3. C)
> - **Graph format:** Adjacency list (A: B, B: C, C: (none))
>
> The adjacency list requires decomposing the chain into explicit pairwise parent-child relationships, testing different structural capacities.
>
> **On formatting vs. causal reasoning deficits.** This is exactly the confound our diagnostics target. The results separate format production difficulty from absent causal competence.
>
> | Model | Numbered (C1) | Arrow (C1) | Semicolon (C3) | Adj. List (C3) |
> |:------|:-------------:|:----------:|:--------------:|:--------------:|
> | LLaVA-NeXT | −0.01 | 0.04 | 0.36 | 0.70 |
> | LLaVA-RLHF | −0.05 | 0.01 | 0.48 | 0.86 |
> | LLaVA-CoT | 0.00 | 0.06 | 0.06 | 0.22 |
> | CogVLM2 | 0.33 | 0.38 | 0.68 | 0.82 |
> | MiniGPT-4 | 0.00 | 0.13 | 0.17 | 0.57 |
> | Qwen3-VL (2025) | −0.01 | 0.01 | 0.01 | 0.22 |
>
> *Table 1: Causal Depth Disparity (CDD) across formats. Values near zero indicate text-structure alignment; high values indicate the plausibility-faithfulness gap.*
>
> Five of six models achieve CDD ≤ 0.01 for numbered format at full scaffolding (C1), indicating causal competence is present when format barriers are removed. The same models show CDD > 0.5 for adjacency lists at minimal scaffolding (C3). If formatting were the sole explanation, all formats would fail similarly; they do not. The gap is format-contingent, not a fixed property of the model.
>
> **Holding format constant isolates model capability.** At minimal scaffolding (C3):
>
> | Model | Numbered | Adjacency List |
> |:------|:--------:|:--------------:|
> | LLaVA-NeXT | 98.3% | 2.7% |
> | Qwen3-VL (2025) | 97.8% | 44.2% |
> | CogVLM2 | 84.4% | 7.2% |
>
> *Table 2: Format compliance rates at minimal scaffolding (C3).*
>
> All three models achieve high compliance for numbered format, confirming they can follow structural output instructions. For adjacency lists under identical conditions, compliance ranges from 2.7% to 44.2%. If the adjacency list format were uniformly "too hard," all models would fail similarly. The variance across models indicates differences in structural capability, not uniform format difficulty.
>
> **On scaffolding sensitivity as an independent dimension.** Even when format compliance is high, scaffolding sensitivity persists. Qwen3-VL achieves 97-98% format compliance for all linearized formats at minimal scaffolding, indicating strong instruction-following. For adjacency lists, compliance drops from 98% at full scaffolding (C1) to 44% at minimal scaffolding (C3), and CDD rises from 0.02 to 0.22. Format compliance and scaffolding sensitivity are independent dimensions. High format compliance does not guarantee scaffolding-invariance. This is what our hierarchical framework predicts: H1 (text-chain alignment) can be satisfied while H2 (scaffolding-invariance) fails.
>
> **On model selection.** Table 1 includes three non-LLaVA architectures. Qwen3-VL-8B-Instruct (2025) provides strong evidence for our position: it achieves the best performance among all tested models, and standard evaluation under fixed scaffolding would conclude it possesses robust causal reasoning. Varying scaffolding shows the capability is partially scaffolding-dependent when structural demands increase. The position paper focused on the LLaVA family for a controlled comparison (isolating the training approach from the architecture), but broader testing confirms that the diagnostic framework generalizes.
>
> **On static images and temporal constitution.** We acknowledge that our diagnostics (CDD, SS) test scaffolding-dependence, not temporal constitution directly. The temporal constitution hypothesis motivates why scaffolding-dependence matters: if models possessed genuine temporal understanding, we would expect scaffolding-invariant performance. Our diagnostics provide indirect evidence; direct tests would require video-based evaluation.
>
> The paper critiques what current benchmarks *presuppose*, not what models *lack*. We do not claim that models cannot perform causal reasoning; we argue that benchmarks using static images while claiming to test causal reasoning presuppose temporal constitution without testing it. The static image methodology is the problem we diagnose. We state this in Section 5: "Testing video-language models with the type of diagnostics we illustrate would be valuable future work. Evidence that video training yields scaffolding-invariant causal structure would count against our position."

---

> > ### Author Rebuttal · Reviewer_P3Kr · 2026-04-04
> >
> > Thanks for the detailed responses. The authors addresses my concerns. I will raise my rating accordingly.

---

### Official Review · Reviewer_Ts3P · 2026-03-13

**Significance:** 3
**Argument Clarity:** 3
**Rating:** 5
**Confidence:** 4

**Questions:**

1. In addition to the plausibility-faithfulness gap, there has also been community discussion around the *reasoning-recall gap* in generative AI. LLMs often appear to succeed on reasoning tasks, when they are in fact relying on memorization/approximate retrieval of information from the training distribution. The authors do not appear to discuss this additional source of noise in AI reasoning evaluation. Can the authors address the extent to which the conflation of reasoning and recall intersects with their problem setting?
    - Recent papers that discuss the reasoning-recall gap:
        - *Reasoning elicitation in language models via counterfactual feedback.* Huyuk et al. 2025.
        - *Re-imagine: Symbolic benchmark synthesis for reasoning evaluation.* Xu et al. 2025.
2. One increasingly common approach for preventing the conflation of reasoning and recall is to use dynamic or generative benchmarks, where tasks are randomly sampled with random variable values or scaling complexity. Another (sometimes overlapping) strategy is to use nonsensical variable names, to ensure that the ground truth was never seen in the training data (preventing recall of commonsense information). To what extent do the authors think such strategies could also help to reveal scaffolding sensitivity / dependence or lack of temporal understanding?
    - Some recent examples:
        - *Re-imagine: Symbolic benchmark synthesis for reasoning evaluation.* Xu et al. 2025.
        - *PhantomWiki: On-Demand Datasets for Reasoning and Retrieval Evaluation.* Gong et al. 2025.
        - *GSM-Symbolic: Understanding the Limitations of Mathematical Reasoning in Large Language Models.* Mirzadeh et al. 2025.
        - Nonsense variable names: *CLADDER: Assessing Causal Reasoning in Language Models.* Jin et al.
        - Nonsense variable names: *Compositional Causal Reasoning Evaluation in Language Models.* Maasch et al. 2025.

**Alternative Views Section:**

Yes

**Compliance With Llm Reviewing Policy A Conservative:**

Affirmed.

**Discussion Potential:**

3

**Final Justification:**

The authors have addressed my questions and I retain my score of accept, on the grounds outlined in the "strengths" section of my original review.

**Paper Summary:**

This position argues that causal reasoning evaluations for VLMs tend to (1) assume understanding of the temporal laws of causation without directly testing for this understanding and (2) conflate performance gains from symbolic scaffolding with intrinsic model capabilities. Preliminary empirical results demonstrate systematic disparities between the fluency of causal text and the external validity of the implied causal structures. The authors advocate against the current over-emphasis on final output accuracy in evaluation, and advocate in favor of benchmarks that directly test for temporal understanding and scaffolding invariance.

**Position:**

Yes

**Position In Title:**

Yes

**Related Work:**

2

**Strengths And Weaknesses:**

# Strengths

1. The authors make a strong argument for more thoughtful benchmark design. The vast majority of reasoning benchmarks today are designed for measuring output accuracy, not the quality of the reasoning process. This output-based view is increasingly facing scrutiny, especially wrt construct validity. The *plausibility-faithfulness gap* is a useful encapsulation of a subproblem that arises in this setting. The concluding observation that "benchmark leaderboards may be tracking scaffolding optimization, not causal reasoning capability" would be a substantial problem if true. In this sense, this position is well-motivated and timely.
2. The authors introduce useful metrics and diagnostics for the problems that they outline.
3. Preliminary results are an insightful first step toward further illuminating this set of problems.
4. The authors provide a comprehensive discussion of alternative views.

---

# Weaknesses & Comments

1. While this position highlights a legitimate set of problems in VLM evaluation, this is a special case of a general problem in AI evaluation. This could be more clearly situation in the broader literature. For example, the output-based view of evaluation, the plausibility-faithfulness gap, and construct validity are general problems in AI evaluation, not local to VLMs and causal reasoning. Chollet discusses the output-based view. Several strong (uncited) position papers have been released on general problems in LLM evaluation, including (among others):
    - *What will it take to fix benchmarking in natural language understanding?* Bowman et al. 2021.
    - *Position: Benchmarking is Broken -- Don't Let AI be its Own Judge.* Cheng et al. 2025.
    - *Position: Evaluating Generative AI Systems Is a Social Science Measurement Challenge.* Wallach et al. 2025.
    - *AI and the everything in the whole wide world benchmark.* Raji et al. 2021.
2. The authors miss several key citations in causal reasoning evaluation, including *CLADDER: Assessing Causal Reasoning in Language Models* (Jin et al.).

**Support:**

3

---

> ### Author Rebuttal · Authors · 2026-03-30
>
> We thank the reviewer for the supportive assessment and the constructive suggestions for strengthening the scholarly context.
>
> **On situating within broader AI evaluation literature.** The reviewer notes connections to general problems in AI evaluation (Bowman et al. 2021, Cheng et al. 2025, Wallach et al. 2025, Raji et al. 2021, Chollet). We will integrate these citations in the camera-ready version.
>
> The general benchmarking critique concerns output-based evaluation across AI tasks. Our position addresses a more specific problem: VLM causal reasoning requires temporal constitution (representing time as the medium through which causes produce effects), and current benchmarks neither test for this prerequisite nor control for scaffolding dependence. This specificity is a strength, not a limitation. The temporal dimension of causal cognition creates evaluation challenges that do not arise for other reasoning tasks. Our diagnostics (CDD, scaffolding sensitivity) operationalize tests for this specific capacity.
>
> We see our work as complementing the general literature by providing domain-specific metrics rather than a derivative special case.
>
> **On the reasoning-recall gap.** This concern complements our framework. We see three distinct dimensions that current benchmarks conflate:
>
> | Dimension | What it detects | Diagnostic strategy |
> |-----------|-----------------|---------------------|
> | Reasoning vs. recall | Memorization vs. genuine reasoning | Novel scenarios, counterfactual perturbation |
> | Scaffolding-dependence | External support vs. internalized capability | Scaffolding variation (our CDD, SS metrics) |
> | Extracted vs. constructed | Pattern matching vs. temporal-causal modeling | Temporal constitution tests (our framework) |
>
> The reasoning-recall literature (Huyuk et al. 2025, Xu et al. 2025) addresses the first dimension. Our position paper contributes diagnostics for the second dimension (scaffolding sensitivity metrics) and the theoretical framework for the third (temporal constitution hypothesis).
>
> These dimensions are orthogonal. A model could pass reasoning-recall tests (genuinely reasoning, not recalling) while still failing scaffolding-invariance tests (reasoning depends on external symbolic support). Conversely, a model could exhibit scaffolding invariance while still relying on memorized patterns. A model could genuinely reason, be scaffolding-invariant, and still extract associations from static features without constructing temporal-causal models. Current benchmarks conflate all three. Our framework disentangles the second and motivates future diagnostics for the third.
>
> **On dynamic/generative benchmarks.** This suggestion aligns with our Call to Action. Dynamic benchmarks with randomly sampled values and nonsensical variable names address reasoning-recall concerns. Combining these strategies with scaffolding variation would address multiple dimensions simultaneously:
>
> - **Reasoning-recall:** Random values and nonsensical names prevent the retrieval of memorized solutions
> - **Scaffolding-dependence:** Varying scaffolding level measures whether performance depends on external support
>
> A benchmark that varies both surface features and scaffolding level would detect both confounds. We will expand Section 4 to note this complementary approach and cite Re-imagine, PhantomWiki, GSM-Symbolic, CLADDER, and Maasch et al.
>
> CLADDER's nonsense variable strategy (Jin et al.) is particularly relevant. By using variables such as "foo causes bar," CLADDER prevents models from relying on commonsense associations, isolating formal causal structure. Our scaffolding variation serves an analogous function: by removing worked examples and output templates, we isolate whether models have internalized causal chain construction or depend on external support. A benchmark combining both strategies would control for reasoning-recall and scaffolding-dependence confounds simultaneously, providing stronger evidence of genuine causal reasoning capability.
>
> **Additional empirical evidence.** We have conducted experiments on 6 VLMs across four structural formats (three linearized chain notations and one graph format) and three scaffolding levels. The additional models include CogVLM2 (2024), MiniGPT-4 (2023), and Qwen3-VL-8B-Instruct (2025):
>
> | Model | CDD (Numbered, C1) | CDD (Adj. List, C3) |
> |:------|:------------------:|:-------------------:|
> | LLaVA-NeXT | −0.01 | 0.70 |
> | Qwen3-VL (2025) | −0.01 | 0.22 |
> | CogVLM2 | 0.33 | 0.82 |
>
> Even Qwen3-VL, the strongest model tested, exhibits scaffolding dependence for structurally complex outputs (adjacency list compliance: 98% at C1, 44% at C3). This addresses the second dimension in our framework: models with near-zero CDD at full scaffolding still show elevated CDD when scaffolding is removed, indicating scaffolding-dependent reasoning rather than internalized capability. The patterns hold across architecture families. Full results appear in our response to Reviewer P3Kr.

---

> > ### Author Rebuttal · Reviewer_Ts3P · 2026-03-31
> >
> > The authors have addressed my questions and I retain my score of accept.

---

### Decision · Program_Chairs · 2026-04-30

**Decision:**

Accept (spotlight)

**Comment:**

All reviewers rated this paper positively (two accept and two borderline accept).

The paper presents a position that is often overlooked in VLM evaluation, which states that we should examine whether VLMs really have the correct causal chain structure during reasoning, instead of simply presuming that VLMs already know about temporal constitution. Therefore, it can inspire in-depth discussions and debates on whether we should focus on this issue, and how we are going to properly address and evaluate it.

The stated position is supported by the preliminary experimental analysis on the LLaVA family. There is evidence supporting both hypothesis for the current reasoning status of VLMs, followed by analysis on the results to explain how scaffolding functions in different VLMs.

The calls to action and suggestions for benchmark developers and VLM researchers are detailed and insightful, which can provide guidance for readers to see which directions to go for to make progress on causal reasoning of VLMs.

The authors make a strong argument for more thoughtful benchmark design. The vast majority of reasoning benchmarks today are designed for measuring output accuracy, not the quality of the reasoning process. This output-based view is increasingly facing scrutiny, especially wrt construct validity. The dangerous plausibility-faithfulness gap is a useful encapsulation of a subproblem that arises in this setting, and the paper effectively explained why current models can generate fluent, convincing narratives that actually lack valid underlying causal structures. The concluding observation that "benchmark leaderboards may be tracking scaffolding optimization, not causal reasoning capability" would be a substantial problem if true. In this sense, this position is well-motivated and timely.

By drawing on interdisciplinary concepts from art, philosophy, and psychoanalysis, the authors propose a highly original diagnostic methodology that moves beyond superficial accuracy metrics to deeply probe internalized model capabilities.

The rebuttal process led to a fruitful discussion of various issues with the paper, which generally satisfied the reviewers.